# Females with Eating Disorders and Urinary Incontinence: A Psychoanalytic Perspective

**DOI:** 10.3390/ijerph19084874

**Published:** 2022-04-17

**Authors:** Qin Xiang Ng, Yu Liang Lim, Wayren Loke, Kuan Tsee Chee, Donovan Yutong Lim

**Affiliations:** 1MOH Holdings Pte Ltd., 1 Maritime Square, Singapore 099253, Singapore; yulianglim95@gmail.com (Y.L.L.); wayren.loke@mohh.com.sg (W.L.); 2Institute of Mental Health, 10 Buangkok View, Buangkok Green, Medical Park, Singapore 539747, Singapore; kuan_tsee_chee@imh.com.sg (K.T.C.); donovan_lim@imh.com.sg (D.Y.L.)

**Keywords:** eating disorders, anorexia, urinary symptoms, functional disorders, incontinence

## Abstract

Eating disorders (EDs) are complicated mental illnesses with significant treatment resistance and dropout rates. For successful treatment, it is important for clinicians to better understand the patients’ narrative and their lived experiences. A thorough psychodynamic understanding of patients’ childhood attachment and primary relationships, personality traits and mental processes is, therefore, crucial for managing patients with ED. Interestingly, several studies have observed an association between functional urinary symptoms and individuals with ED. EDs such as anorexia nervosa are associated with an increased risk of all urinary symptoms, and functional incontinence was also more common in extreme female athletes with low energy availability and with disordered eating. There is, however, a dearth of literature describing this relationship, and the underlying mechanisms remain remote. In this paper, we present a psychoanalytic approach to the presence of urinary symptoms in females with EDs. We hypothesize that these symptoms are tied to specific traits or characteristics of ED patients, namely the overarching need for control, a pathological strive for perfection and the self-denial of basic bodily urges. This is discussed in relation to psychopathological processes, development and personality factors commonly seen in patients with ED.

## 1. Introduction

Eating disorders (EDs), including anorexia nervosa (AN) and bulimia nervosa (BN), are a group of mental disorders with complex biological and environmental etiopathologic factors [1]. Despite the advent of modern psychopharmacology, these disorders remain extremely challenging to manage as patients often display ego-syntonic features and significant dropout and resistance to treatments [2]. The holistic treatment of ED patients, therefore, involves a good psychodynamic understanding of their childhood attachment and primary relationships, personality traits and inner mental landscape. 

Interestingly, several studies have highlighted an increased prevalence of functional urinary disorders in females with low energy availability [3] and those with disordered eating or diagnosed EDs [4,5]. Possible urinary symptoms include urgency, frequency and incontinence. This observed association could be more than fortuitous—it may be a manifestation of their underlying personality traits of perfectionism and the need for control.

There is a paucity of literature describing this relationship between EDs in females and functional urinary disorders. In this paper, we present a psychoanalytic approach to the presence of urinary symptoms in females with diagnosed ED in relation to psychopathological processes, and developmental and personality factors. It is hoped that a more thorough understanding of the underlying mechanisms can be achieved.

## 2. Psychopathological Processes Underlying EDs

The biopsychosocial pathogenesis of ED is widely acknowledged [1]. In ED, there is a persistent overevaluation of body shape and weight [6]. As a result, individuals have extreme weight-control behaviors, including dietary restraint, purging, abuse of laxative and diuretic substances and over-exercising, and the ability to control their weight underlies the core psychopathology of ED [7]. Consistent with the prevailing Fairburn’s transdiagnostic cognitive-behavioral model of ED, this need for control permeates all domains of life and loss of control in one domain may reinforce controlling behaviors in other domains [8]. This implies that an individual’s perceived sense of powerlessness in response to life’s stressors and a lack of control will exacerbate ED symptoms. 

Qualitative studies into the lived experience of patients with ED lend support to the notion that a general perceived lack of control is a significant contributing factor to the development and persistence of ED [9,10]. Patients typically recall starting to restrict food intake and watch their weight at a time when they felt hopeless, powerless and their lives to be chaotic and spiraling out of control. Controlling one’s diet and weight was then thought to be the “solution”. As aptly described by a patient with ED, “When I started changing my eating habits it was… because I didn’t feel in control of my life or of myself. Controlling what I ate was one way of controlling at least part of my life… I felt that if I could control what went in and out and how much exercise I did then I could control other things in my life” [11]. Others have also communicated similar experiences, where “In a world where it’s not possible to be perfect at everything, ED had promised that I could always be perfect at one thing” [12]. In EDs, there seems to be an overarching need for control, in the background struggle for a sense of identity and for competence. The precursor to disordered eating is a perception of “loss of control”, or poor coping in a diathesis-stress model [13], and ED behaviors are seen as a means of regaining that control, and a maladaptive marker of self-efficacy [14].

ED symptomatology and resultant behaviors could thus be understood as a frantic attempt to compensate for a deep vein of inferiority and a lack of control experienced in other domains of the individual’s life. Besides restricting food intake and drinking, other controlling behaviors, including denying bodily needs, such as urination, are possible and related to this inherent drive. The perceived gain in control (and, by extension, autonomy and competence) allows individuals to (partly) negate the negative affect associated with general life dissatisfaction and interpersonal problems [15]. 

Several lines of evidence have also characterized EDs as developmental disorders, with insecure attachment in early childhood, early maladaptive schemas and other biopsychosocial factors that hamper the mature development of the self [16,17,18]. The disorder mirrors the internal conflict and ambivalence surrounding the self and the “ED identity” [19]. Individuals attach their identity to their illness and fear losing control on eating and other aspects of their lives. Psychotherapy helps individuals with ED engender a healthier sense of self and acknowledge their complex, albeit realistic emotional and physical needs [19]. 

## 3. Psychoanalytic Perspectives

Historically, Freud and other psychoanalysts have also viewed bladder and bowel control as a pivotal step in the development of the child and his or her role as a competent adult [20]. By extension, the development of secondary enuresis or encopresis in adulthood is thought to be associated with a problem in the character or personality of the individual [21] or the result of emotional or physical trauma in childhood [22]. Studies do indicate an increased prevalence of sexual trauma amongst individuals with ED [22]. The emotional dysregulation in individuals with ED has also been linked to body dissatisfaction, sexual dysfunction and the fear of future sexual trauma [23]. The development of incontinence could also be an age regression or unconscious defense mechanism so that these individuals cannot be sexualized.

Related to this, although literally starving and harming their physical health, some patients with ED are strongly repulsed by food due to their intense fears of eating and fantasies of oral impregnation [24]. From a psychoanalytic point of view, the act of feeding oneself or the state of being fat bears a deeply disturbing symbolic significance for these patients. Psychoanalysts suggest that “oral impregnation” fears stem from fixation at the oral stage of psychosexual development [20,24].

The idea of attachment issues in childhood and a dysfunctional self is also advanced by the ego psychological understanding of ED [25] and interpersonal theory [26]. The ED symptomology is thought to be rooted in an early disordered mother–child relationship as feeding symbolically represents the nurturance and lack thereof from one’s mother [25]. A psychotic ego organization then produces the obsessional thinking and behaviors (e.g., overfocus on feelings of loss of control and an intense need to regain that control, or the desperate attempts to maintain some idiosyncratic perfection) seen in certain ED patients.

## 4. Confluence of Personality Traits

Perfectionism and asceticism are also personality traits commonly present in individuals with EDs [27,28]. These personality characteristics are sometimes regarded as a trait or a symptom or can also be understood as a process. Perfectionism—the tendency to hold onto unrelenting and unrealistically high standards—has long been implicated in the development and maintenance of EDs [29]. The constant obsession with one’s body weight and image and overdependence on self-evaluation clearly play a role in the etiology and maintenance of EDs. Likewise, asceticism in ED patients is thought to be related to the underlying need for control [30]. Historically, asceticism is interpreted in the spiritual or religious context whereby persons engage in pious self-denial in the pursuit of virtuous perfection [27]. In the context of EDs, asceticism refers to an individual’s predisposition to punishing self-denial and self-sacrifice (at the expense of one’s health) and the exertion of complete control over his or her bodily needs.

These personality traits at least partly influence ED clinical behaviors and perpetuate feelings of guilt, self-loathing and denial. As personality affects thoughts, emotions and behavior [31], patients with ED may engage in certain behaviors to reinforce their own perfectionistic and ascetic beliefs. This includes denying the conscious need to urinate (a bodily function). Moreover, individuals with perfectionism respond to the failure to meet their demanding standards with further self-criticism and self-punishment [29].

As noted by previous studies, there may also be abnormal interoceptive awareness in individuals with ED [32]. Poor interoceptive sensitivity means that these individuals may have an impaired awareness of their feelings or emotions and poorer emotional regulation. This may also affect the perception of stimuli arising within the body. Females with incontinence issues have been found to manifest abnormal activation of portions of the brain that govern interoception, the perception and interpretation of physiologic bodily stimuli [33]. These abnormalities may modulate urge perception and abnormal storage in the genesis or persistence of urinary symptoms.

## 5. Association with Functional Urinary Symptoms

Functional urinary disorders consist of a diverse group of disorders for which there is no apparent neurological or structural abnormality [33]. One study has shown that adolescents with AN were associated with an increased risk of all urinary symptoms [34]. Athletes with disordered eating have also been found to be three times more likely to have urinary incontinence symptoms than those without disordered eating [4].

There are many causes of urinary incontinence, and they can be broadly classified as stress incontinence, urge incontinence, overflow incontinence, mixed incontinence or functional incontinence [35]. In terms of biological mechanisms, in patients with ED, the chronic starvation and malnutrition states may weaken pelvic floor muscles and the ligaments and fascia that support the pelvic floor [36]. Repeated self-induced vomiting may also further weaken the pelvic floor as it generates increased intra-abdominal pressure. Lower levels of estrogen, which are common in AN, are also risk factors for both stress and urgency urinary incontinence [37]. Another potential contributory factor includes concomitant psychotropic medications, as selective serotonin reuptake inhibitor (SSRI) antidepressants [38], antipsychotics [39] and even benzodiazepines [40] have all been linked to urinary incontinence. They may have anticholinergic effects or agonist effects on 5-HT4 receptors, which are also present in the lower urinary tract and could evoke responses in the bladder and urethra.

Herein, we present another potential mechanism for functional incontinence in these individuals. Functional incontinence may arise due to non-genitourinary factors, such as cognitive, psychological or physical impairments, which affect one’s ability to void independently [41]. In patients with ED, there may be delayed or abnormal voiding of the bladder due to conscious control, and functional urinary incontinence may thus result.

As aforementioned, this could be tied to the underlying need for control, decreased awareness of bodily stimuli or the reinforcement of ascetic beliefs or other overvalued ideas held by these individuals (Figure 1). Denying oneself of otherwise normal bodily function is in keeping with the inner driving forces of ED. Urinary incontinence in ED may also share psychoanalytic roots with “dietary incontinence”. The latter manifests as binging and purging behaviors. These are all slip-ups in the unrelenting quest for control and perfection, which feeds on itself, fueled by a sense of guilt and self-loathing [42], similar to a psychodynamic positive feedback loop.

For clinicians, urodynamic studies are useful in assessing these symptoms. They allow for the assessment of the bladder function throughout the process of urinary storage and voiding, thus allowing us to pinpoint specific pathologies such as an overactive detrusor causing urge symptoms or an impaired bladder sensation causing delayed voiding [43]. Urodynamic studies could be an empirical method to correlate psychopathology with functional biological pathology and to gain a greater understanding of this phenomenon.

## 6. Conclusions

ED is a complex psychiatric disorder, and it is important for clinicians to better understand patients’ narrative and their psyche. A vicious cycle of control and the fear of losing control is apparent, and we postulate that functional urinary symptoms may be another manifestation of their underlying idiosyncratic need for control based on psychoanalytic theory in this patient group. The manifested urinary incontinence may also be a reflection of adverse early life experiences. Decreased interoceptive awareness may play a role as well. Functional urinary symptoms should be carefully considered when approaching patients with ED. There is certainly much to be learned vis-à-vis the profound meaning and nature of these disorders.

## Figures and Tables

**Figure 1 ijerph-19-04874-f001:**
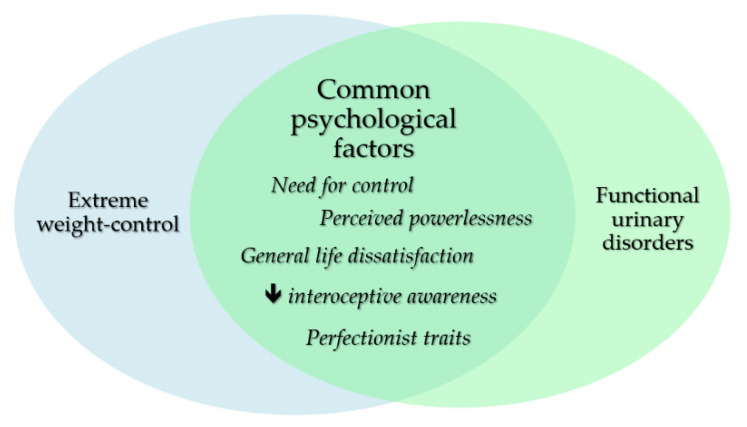
Psychological factors implicated in functional urinary disorders among females with ED.

## Data Availability

No data were generated during the study.

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
