# Peer review of "Females with Eating Disorders and Urinary Incontinence: A Psychoanalytic Perspective"

_ijerph, 2022, doi:10.3390/ijerph19084874_

Round 1

Reviewer 1 Report

The paper presents a conceptual and explicative hypothesis about the functional meaning of urinary incontinence in female patients suffering from eating disorders, in the framework of a psychodynamic reading of ED.

The hypothesis is well articulated in principle, and discussed with respect to indirect evidence derived from clinical and empirical studies already published in the literature in related contextes.

In order to make the paper suitable for pubblication, the authors should make an effort to describe possible empirical  methods that could provide more direct evidence to support their hypothesis.

Please revise the English in the Abstract.

Author Response

  1. We have made edits to the abstract to make it more readily accessible to the general reader.
  2. We have added to the discussion section, "For clinicians, urodynamic studies are useful in assessing these symptoms. They allow for assessment of the bladder function throughout the process of urinary storage and voiding, thus allowing us to pinpoint specific pathologies such as an overactive detrusor causing urge symptoms, or an impaired bladder sensation causing delayed voiding [36]. Urodynamic studies could be an empirical method to correlate psychopathology with functional biological pathology and to gain a greater understanding of this phenomenon."

Reviewer 2 Report

Abstract: It is necessary to better explain the conclusions, from line 20 what is expressed is unclear even for a reader with psychodynamic training.

Introduction: line 37: as for the abstract, explaining this concept better would be incomprehensible to a reader of a different psychological orientation

Conclusions: expand the conclusions by taking up in an organic but summarized way the concepts dealt with in the different paragraphs in relation to the specific topic of urinary problems in patients with eating problems

Author Response

  1. We have made edits to the abstract to improve its clarity of expression, "We hypothesize that these symptoms are tied to specific traits or characteristics of ED patients, namely the overarching need for control, a pathological strive for perfection, and the self-denial of basic bodily urges."
  2. Line 37 now reads, "Possible urinary symptoms include urgency, frequency and incontinence."
  3. The conclusion now reads, "ED is a complex psychiatric disorder and it is important for clinicians to better understand patients’ narrative and their psyche. A vicious cycle of control and the fear of losing control is apparent, and functional urinary symptoms may be another manifestation of their underlying idiosyncratic need for control. The manifested urinary incontinence may also be a reflection of adverse early life experiences. Decreased interoceptive awareness may play a role as well. Functional urinary symptoms should be carefully considered when approaching patients with ED. There is certainly much to be learnt vis-à-vis the profound meaning and nature of these disorders."